# Pregnancy Outcomes after Frozen Embryo Transfer and Fresh Embryo Transfer in Women of Advanced Maternal Age: Single-Center Experience

**DOI:** 10.3390/jcm11216395

**Published:** 2022-10-28

**Authors:** Yao Chen, Jianbo Zhou, Yandong Chen, Jihong Yang, Yingying Hao, Ting Feng, Ruizhi Feng, Yun Qian

**Affiliations:** 1Reproductive Center, Second Affiliated Hospital, Nanjing Medical University, Nanjing 210011, China; 2Department of Obstetrics and Gynecology, Binhai County People’s Hospital, Yancheng 224000, China; 3State Key Laboratory of Reproductive Medicine, Nanjing Medical University, Nanjing 211166, China; 4Second Affiliated Hospital, Nanjing Medical University, Nanjing 210011, China

**Keywords:** advanced maternal age, frozen embryo transfer, fresh embryo transfer, pregnancy outcomes

## Abstract

Delayed childbearing leads to increased assisted reproductive technology use by women of advanced maternal age (AMA). It is unclear whether fresh or frozen embryo transfer (FET) is the better option. We aimed to assess maternal and neonatal outcomes in patients having their first FET after a freeze-all cycle versus those having their first fresh embryo transfer (ET). We reviewed 720 women of AMA undergoing a first fresh ET (n = 375) or FET (n = 345) between January 2016 and April 2021. No significant difference in the live birth rate was found between FET and fresh ET (19.7% vs. 24.3%, *p* = 0.141). The clinical pregnancy rate was significantly lower in the FET group than in the fresh ET group (26.4 % (91/345) vs. 33.6% (126/375), *p* = 0.035), but FET resulted in higher birthweights (3217.16 ± 734.44 vs. 3003.37 ± 635.00, *p* = 0.037) and was associated with a lower incidence of preterm births (2.6% vs. 5.6%, *p* = 0.046). The risks of other maternal and neonatal outcomes did not differ significantly between the groups. Among women of AMA, the transfer of frozen embryos did not result in significantly higher rates of live birth than fresh embryos did; however, a freeze-all strategy may not be beneficial for the women of AMA.

## 1. Introduction

Advanced maternal age (AMA) is a critical social and clinical issue [1]. The proportion of women who delay childbearing has increased substantially, possibly related to increased education and female employment, career goals, efficient contraceptive strategies, and the widespread and misleading idea that in vitro fertilization (IVF) can compensate for the natural decline in fertility with age [2,3,4].

Female fertility declines significantly after age 35 and more rapidly after age 40 [5,6]. AMA is associated with a decline in ovarian reserve and oocyte competence [1]. Overcoming these difficulties is a challenge for physicians. No remedies are currently available to counteract the aging-related fertility decline. However, different therapeutic approaches can be offered to women >35 years undergoing IVF.

Several studies have compared the outcomes of frozen embryo transfer (FET) and fresh embryo transfer (ET) [7,8,9,10,11,12]. Some randomized trials suggest that freezing all embryos in a fresh IVF cycle followed by frozen–thawed ET in subsequent cycles might improve pregnancy and live birth rates [7,8,9]. FET may provide a more favorable intrauterine environment for embryo implantation and placentation by avoiding the supra-physiologic hormonal levels after ovarian stimulation. However, FET is associated with increased risks of macrosomia, large size for gestational age, and several maternal complications [10,13,14]. Women assigned to FET also have a lower rate of ovarian hyperstimulation syndrome but a higher rate of preeclampsia [8]. Few studies have compared pregnancy outcomes and maternal complications in AMA women treated with FET and fresh ET. Questions remain about whether FET could improve outcomes in AMA women. This retrospective study aims to compare the effectiveness of FET versus fresh ET in women of AMA (≥35 years of age).

## 2. Materials and Methods

### 2.1. Study Design and Participants

The retrospective research was conducted in the Second Affiliated Hospital of Nanjing Medical University between January 2016 and June 2021. The trial was approved by the ethics committees of the Reproductive Center, Second Affiliated Hospital of Nanjing Medical University. In both groups, only women aged 35–50 years were included. Patients who underwent their first cycle of FET after a freeze-all cycle and those who underwent their first cycle of fresh ET were included and compared. Freeze-all strategy was implemented in the case of high progesterone concentration on the trigger day (>1.5 ng/mL) or to prevent OHSS. Due to the supra-physiologic hormonal levels, patients implemented the strategy. Freezing was also performed in the case of unresected hydrosalpinx before IVF, unexplained uterine effusion, undiagnosed endometrial polyps, submucosal fibroids, thin endometrium thickness, or unexplained fever or acute infection. The treatment of the patients depends on the professional judgment of the experts, and the personal wishes of the patients should also be considered. Patients who needed oocyte donors, patients with endocrine disorders (hypothyroidism, hyperthyroidism, diabetes mellitus, hyperprolactinemia), and patients who previously underwent chemotherapy and radiotherapy were excluded. Patients with serious chronic diseases such as hypertension, symptomatic heart disease, and a history of cardiovascular and cerebrovascular disease were also excluded. Cycles that met the inclusion criteria were extracted from the Assisted Reproductive Technology (ART) database. A flow chart of the included and excluded cycles is displayed in Figure 1.

### 2.2. IVF Treatment

Patients received the ovarian stimulation protocol, including the flexible short protocol or the depot GnRHa protocol based on their ovarian response.

#### 2.2.1. Depot GnRHa Protocol

In the depot GnRHa protocol, patients received a subcutaneous injection of 3.75 mg long-acting triptorelin (Decapeptyl; Ferring, Saint-Prex, Switzerland) on Day 2 of the cycle when ovarian quiescence was confirmed, and endometrial thickness was less than 5 mm on ultrasound. When complete pituitary desensitization was achieved (28 days after triptorelin administration), ovarian stimulation was started with 150–225 IU/day of follicle-stimulating hormone (FSH). The initial and subsequent FSH doses were adjusted according to the patient’s age, body mass index (BMI), antral follicle count (AFC), and follicular growth response. Human chorionic gonadotrophin (HCG) was used as the trigger when more than two follicles reached 18 mm in diameter.

#### 2.2.2. Flexible Short Protocol

The flexible short protocol was performed according to our previously published standard protocols [15]. All patients were called for follow-up on the third day of the menstrual cycle, and 0.05 mg/d Triptorelin (Decapeptyl; Ferring, Saint-Prex, Switzerland) was injected from that day until HCG injection. Gonadotrophin (FSH) was injected when estrogen began to rise, and there was at least one follicle larger than 5 mm in diameter. The dose of gonadotropin ranged from 150 to 300 IU per day, depending on the woman’s age, anti-Müllerian hormone levels, and response to FSH. Follicular development was monitored using ultrasonography and testing of estradiol and progesterone levels. When the mean diameter of at least two leading follicles was 17 mm, HCG (10000 IU) was administered subcutaneously, and oocyte retrieval was performed 36 h later. Based on the number of oocytes retrieved, high responders were defined as 15 or more oocytes retrieved, intermediate responders were defined as 6–14 oocytes retrieved, and low responders were defined as 1–5 oocytes retrieved.

Oocytes were fertilized by either conventional IVF or intracytoplasmic sperm injection (ICSI) according to the results of semen analysis. Evaluation and grading of the embryo were performed according to The Istanbul consensus [16]. Daily progesterone was administered after oocyte retrieval in the fresh ET group. A maximum of two cleavage stage embryos were transferred on Day 3 after oocyte retrieval. With guidance from transabdominal ultrasound and a full bladder, fresh cycle embryos were transferred three days after IVF or ICSI. Luteal support was used for ten weeks after oocytes retrieval.

In the FET group, all the high-quality embryos were frozen by vitrification for cryopreservation on the third day after oocyte retrieval. Vitrification of embryos was carried out by the two-step protocol of Mukaida et al. [17]. Dimethylsulphoxide (DMSO), ethylene glycol and sucrose were used as the cryoprotectant agents. All steps were performed on a heated laminar flow hood at 37 ℃. Embryos were incubated in Vitrification solution1 which was consisted of 7.5% DMSO and 7.5% ethylene glycol for 2 min. Then, the embryos were moved to Vitrification Solution 2 containing 15% DMSO, 15% ethylene glycol, 10 mg/mL Ficoll and 0.65 mol/L sucrose for 35 s. The embryos were quickly loaded onto a cryoloop covered with a thin film of the same cryoprotectant. The cryoloop with embryos was immediately put into a vial filled with liquid nitrogen. All handling of vials containing vitrified embryos was performed while keeping the samples immersed in liquid nitrogen. Then, canes were placed in a liquid nitrogen storage tank.

FET cycles were performed in artificial hormone replacement cycles. Estradiol treatment started on the second or third day of the menstrual cycle with oral estradiol valerate (Bayer, Leverkusen, Germany) at 4 to 8 mg daily. Endometrial thickness was monitored on the 9th to 11th day of the cycle. On the 14th to 15th day, when the endometrial thickness reached 8 mm or more, oral dydrogesterone (Cyclogest, Actavis) at a dose of 20 mg twice daily was added. Warming of vitrified embryos was also performed at 37 ℃. The cryoloop was directly immersed in a warming solution containing 0.25 mol/L sucrose for 2 min. The embryos were visualized using the dissecting microscope. Then, the embryos were moved to the second warming solution, which contained 0.125 mol/L sucrose and incubated for 3 min. The final rinse was in culture media for 5 min and incubated at 37 °C with 5.5% CO_2_. The warmed embryos were cultured in Global Blastocyst Medium with 10% SSS until transfer. Morphological survival assessment was taken immediately upon warming. The measure of cryopreservation efficiency is the intactness of survival. An embryo is considered to have survived if 50% or more of the original cell number remains intact after thawing/warming per the Alpha Consensus Meeting on Cryopreservation, 2012 [18]. Only embryos that met the morphological survival assessment could be transferred. A maximum of two high-quality cleavage stage embryos were thawed on the day of ET, three days after the start of progesterone. Two hours after thawing, frozen embryos were transferred into the uterus under transabdominal ultrasound guidance. Luteal support was also provided for 10 weeks after FET. A pregnancy test was performed 14 days after the ET by measuring HCG in the blood. An HCG value >5 U/mL was considered positive. If conception occurred, transvaginal ultrasound was performed to confirm the presence of an intrauterine gestational sac three weeks later. Biochemical pregnancy was defined as an HCG value >5 U/mL and no gestational sac on transvaginal ultrasound. Clinical pregnancy was defined as the presence of a live fetus and/or a gestational sac in ultrasonography four weeks after embryo transfer. The implantation rate was calculated as the total number of gestational sacs divided by the total number of embryos transferred. The total follow-up duration was 12 months.

### 2.3. Measurement Outcomes

The primary outcome was live birth after the initial transfer, defined as delivery of any neonate ≥28 weeks gestation with a heartbeat and breath. The secondary efficacy outcomes included moderate and severe ovarian hyperstimulation syndrome (OHSS), biochemical pregnancy, implantation, clinical pregnancy, miscarriage, ectopic pregnancy, birth weight, congenital anomaly, and obstetrical and perinatal complications (i.e., gestational diabetes, gestational hypertension, preeclampsia, preterm birth, cesarean section, gestational age, and perinatal mortality). Pregnancy and birth information were obtained using the participants’ medical and obstetric records.

### 2.4. Statistical Analysis

The primary and secondary outcomes were assessed by comparing the outcomes after the first embryo transfer. All statistical analyses were performed using SPSS software (version26.0, IBM SPSS Inc, Chicago, USA). We used the Chi-square or Fisher’s exact test to assess between-group differences in categorical variables and the independent samples t-test to assess differences in continuous variables. The data are shown as the mean ± SD and percentages, respectively. A value of *p* < 0.05 was considered statistically significant.

## 3. Results

### 3.1. Baseline Characteristics of the Study Groups

A total of 345 FET cycles and 375 fresh ET cycles were included from January 2016 through April 2021. The baseline characteristics of the study groups are provided in Table 1. The mean age of the patients was 39.48 ± 3.75 years in the FET group and 39.80 ± 3.72 years in the fresh ET group. There was no significant difference in age, duration of infertility, infertility type, BMI, the number of previous conceptions, mean anti-Müllerian hormone level, antral follicle count (AFC), causes of infertility, and baseline sex hormone among the two groups. Most of the baseline characteristics were comparable, while the antral follicle count in the right ovary was different (5.59 ± 3.05 vs. 6.09 ± 2.73, *p* = 0.022).

### 3.2. Clinical Features of the Study Groups

The fresh ET and FET groups were comparable in the total amount of gonadotropins, days of ovarian stimulation, treatment protocols and endometrial thickness before transfer (Table 2). On the day that oocyte maturation was triggered, there were no significant differences in estradiol and progesterone levels between the two groups (Table 2). Both groups had a comparable number of oocytes retrieved, a comparable number of embryos transferred, and a comparable number of good quality embryos. Insemination modes also did not differ significantly between the two groups.

### 3.3. Fertility and Neonatal Outcomes of the Study Groups

As shown in Table 3, there was no significant difference in the rate of live birth between the FET group and the fresh ET group (19.7% vs. 24.3%, *p* = 0.141). The rate of biochemical pregnancy, implantation rate, ectopic pregnancy, and miscarriage also did not differ significantly between the two groups (Table 3). In addition, the clinical pregnancy was 26.4% (91/345) in the FET group and 33.6% (126/375) in the fresh ET group, with a statistically significant difference. 

FET resulted in a higher birthweight than fresh ET (3217.16 ± 734.44 vs. 3003.37 ± 635.00, *p* = 0.037) (Table 3). However, there was no significant difference in the incidence of macrosomia and low birth weight between the two groups. The risks of congenital anomalies were also similar between the two groups.

### 3.4. Maternal Outcomes of the Study Groups

The risk of moderate or severe ovarian hyperstimulation syndrome did not differ significantly between the two groups (Table 4). There were no significant differences in the incidence of maternal outcomes, such as gestational hypertension, preeclampsia, HELLP syndrome, gestational diabetes, preterm premature rupture of the membranes (PPROM), and cesarean section, except for the rate of preterm birth. There was no perinatal mortality in both groups.

## 4. Discussion

This retrospective study compared the fertility, perinatal, and maternal outcomes after FET and fresh ET cycles in women of AMA. To reduce heterogeneity, we retrospectively analyzed only the first fresh transfer with the first frozen transfer after a first freeze-all cycle; only transfers of cleavage-stage embryos were included. Embryos were evaluated according to the Istanbul consensus [16], and good embryos were defined as grade I, cell number of 7 to 9, even cell size, less than 10% fragmentation, and no multinucleation. The average number of good-quality embryos in FET group was 1.9, while that in ET group was 1.92. There were no statistically significant differences between the groups in the number of good-quality embryos. Only embryos that met the morphological survival assessment could be transferred. A maximum of two grade 1 or 2 embryos were transferred into the uterus under ultrasonographic guidance. Our study used either the flexible short protocol or the depot GnRHa protocol for ovarian stimulation. There was no statistical difference in the proportion of flexible short or depot GnRHa protocols between the two groups. The distribution of causes of infertility, age, BMI, and hormone levels on the day of triggering of oocyte maturation, the total amount of gonadotropins, days of ovarian stimulation, and endometrial thickness before transfer were similar, while the antral follicle count in the right ovary was different. The significantly different number of antral follicle counts may be due to the measurements made by different B-ultrasound doctors. The AMH and baseline sex hormone did not differ significantly.

Freeze-all embryos were vitrified at d3 post insemination. The implementation of vitrification made IVF safer and more efficient. The most important source of damage during cryopreservation was ice crystal formation, including extracellular and especially intracellular. When vitrification was performed correctly, ice crystal formation is eliminated in both the intracellular and extracellular spaces [19]. A measure of cryopreservation efficiency is the intactness of survival. Vitrification ensures a very high rate of survival (typically around 95% or above) of embryos, independent of what stage they were frozen. Another outcome measure of cryopreservation efficiency was the ability of an embryo to continue sequential development and then implant viability. Improved viability is directly reflected in the improved implantation rates. In some studies, vitrification also allows embryos to maintain high rates of viability with implantation rates similar to the rates of fresh embryos [20,21,22].

In earlier studies, FET showed a higher implantation rate, clinical pregnancy rate, and live birth rate than ET [23,24]. However, two recent randomized controlled trials indicated that FET did not significantly increase the live birth rate [11,25]. Women with PCOS treated with FET showed a higher live birth rate; we speculate that it may be due to the adverse uterine environment after fresh ET [8]. A recent meta-analysis reported a higher rate of live births after the first frozen ET compared with that after the first fresh ET in high responders but not in normal responders [26]. In an observational study, in women who retrieved 10–15 oocytes, FET had higher rates of implantation and clinical pregnancy than fresh ET, but not in women who retrieved 4–15 oocytes [27]. Based on the number of oocytes extracted, the fresh ET group’s clinical pregnancy and live birth rates were higher than the FET group in normal and low responders [28]. FET may be beneficial in women who are high responders to ovarian stimulation or are at a high risk of OHSS, but it appears to decrease the good outcomes in low or intermediate responders [28].

We found that the rates of live birth and implantation were comparable between the two groups. According to the number of oocytes retrieved (8.12 ± 6.78 vs. 8.16 ± 4.79), most of our study patients were in intermediate or low ovarian response. Consistent with previous results, our study confirmed that the ET group had a higher clinical pregnancy rate than the FET group. These findings suggest that FET may not benefit women of AMA. Furthermore, some patients with low oocyte yield were treated with FET, which might have driven our low pregnancy rates in the FET group. This may explain some of the differences between our results and other recently published randomized controlled trials that reported the benefits of FET primarily in patients with high response.

### 4.1. Neonatal Outcomes

Multiple studies have shown that birth weights associated with FET are higher than those associated with fresh ET [29,30,31]. As shown in multiple previous studies, we consistently found higher birthweights in the FET group. However, there were no significant differences in the risk of macrosomia and low birth weight. As demonstrated in two meta-analyses, FET is associated with a decreased risk of low birthweight [32,33]. The incidence of low birthweight was lower with FET than fresh ET, although the between-group difference was not significant. Given the small amount of data available for both groups, larger prospective and randomized studies are needed to validate our findings. No significant between-group difference in the rate of congenital abnormalities was identified in a large cohort study [34]. In our study, there was only one congenital anomaly among live newborns in the FET group. A few studies reported that FET could increase the perinatal mortality rate in singletons [13,35]. However, our study indicated no perinatal mortality in either group. The trial could not accurately analyze differences in these uncommon results, and further studies with a larger sample size in a meta-analysis are required to evaluate these results.

### 4.2. Maternal Outcome

There were no significant differences in maternal outcomes between the two groups, except for the rates of preterm birth.

FET is associated with a lower risk of moderate/severe OHSS in both high and normal responders compared to fresh ET [8,11]. Although OHSS is more common in younger patients, it cannot be ruled out in women of AMA with good ovarian reserves [36]. Compared with previous trials, we did not find a significantly lower rate of OHSS in the FET group than in the fresh ET group. Given the lower average number of oocytes retrieved in women of AMA, the risk of OHSS in both groups was low. Only one patient was diagnosed with moderate OHSS in the fresh ET group.

The risks of preterm birth after FET remain debatable. Two randomized trials involving FET and fresh ET found no significant difference in preterm birth rates [11,25]. In some studies, FET was associated with a decreased risk of low preterm birth compared to the ET group [10,32,33,37]. We also found that FET was related to a lower incidence of preterm birth. Therefore, more evidence is needed to reach a convincing conclusion.

It remains controversial whether FET increases hypertensive disorders of pregnancy compared with fresh ET. A meta-analysis reported that the singletons born after FET were at increased risks of maternal hypertensive disorders of pregnancy than fresh ET [33]. A retrospective analysis in Japan suggested that FET was associated with a higher incidence of hypertensive pregnancy disorders [10]. Moreover, other studies have indicated that FET increased the incidence of preeclampsia [13,32,33]. However, a multicenter randomized trial in ovulatory women was inconsistent with these studies [11]. The present study found no significant difference in the risk of hypertensive disorders of pregnancy between the two groups.

Most randomized trials comparing FET with fresh ET have involved women with an intermediate or high response to ovarian stimulation [7,8,11]. As an elevated estradiol level may adversely affect endometrial receptivity after fresh ET, FET may provide a more physiologic uterine environment [38]. Currently, no relevant randomized controlled trials have assessed low responders, where the risk of OHSS is low. A retrospective analysis at a single center recently provided such data on women of AMA [39].

## 5. Conclusions

We found no significant difference in the live birth rate between FET and fresh ET in women of AMA. Consistent with previous studies, the clinical pregnancies are higher after fresh ET in women of AMA. Most maternal and neonatal outcomes in our study did not differ significantly between the two groups, except for the birth weights and the preterm birth rate. FET was associated with a decreased risk of preterm birth and a higher birthweight than fresh ET. The novel data in our study imply that freeze-all strategies are beneficial in high responders but not in intermediate or low responders and not in women of AMA. Therefore, with the increase in women of AMA, and the increase in the application of FET, this study provides new findings and raises new questions about optimum protocols for ART patients. Individualized treatment should be adopted, and the most appropriate protocol should be chosen based on each patient’s age and ovarian response. Our findings require further randomized studies or multicenter studies for validation.

## Figures and Tables

**Figure 1 jcm-11-06395-f001:**
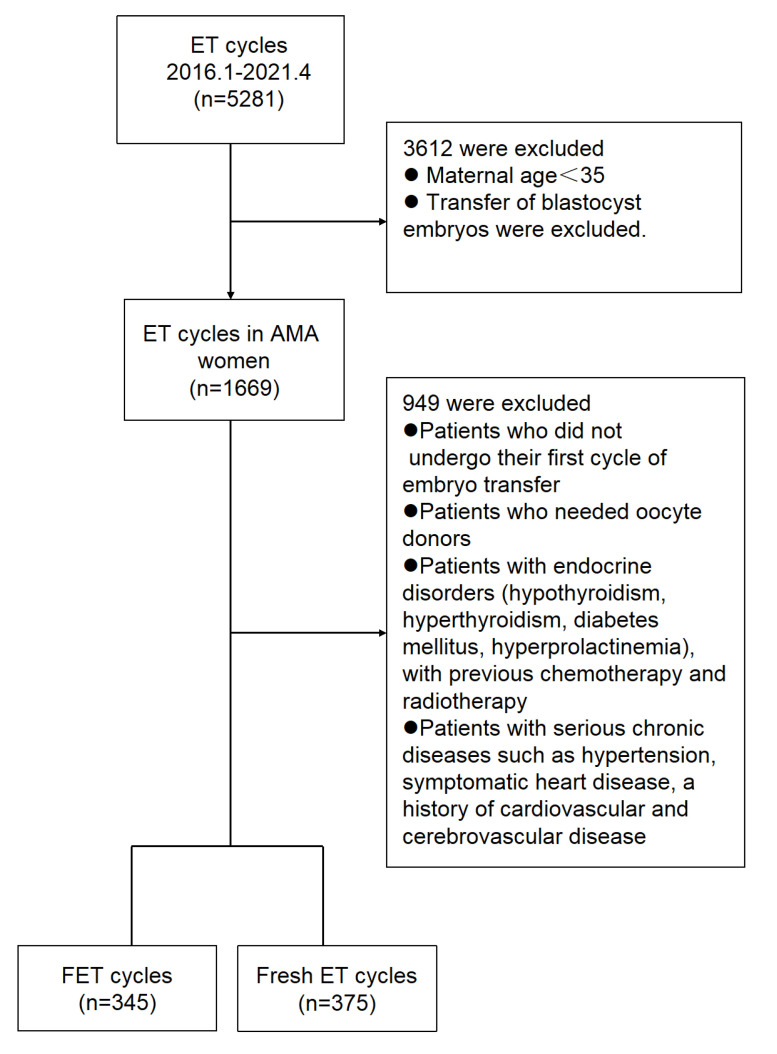
Enrollment and outcomes. Abbreviations: ET—embryo transfer, AMA—advanced maternal age, FET—frozen embryo transfer.

**Table 1 jcm-11-06395-t001:** Baseline characteristics of the FET group and fresh ET group.

Parameters	FET (n = 345)	Fresh ET (n = 375)	*p*-Value
Age, year	39.48 ± 3.75	39.80 ± 3.72	0.258
Duration of infertility, year	4.39 ± 4.06	4.24 ± 3.82	0.633
Infertility Type, n (%)			
Primary	64 (18.6)	74 (19.7)	0.687
Secondary	281 (81.4)	301 (80.3)	0.687
Body mass index	23.19 ± 2.96	23.02 ± 2.96	0.430
Number of previous conceptions	1.87 ± 1.50	1.98 ± 1.58	0.358
Causes of Infertility, n (%)			
Tubal factor	175 (50.7)	174 (46.4)	0.246
Diminished ovarian reserve	162 (47.0)	199 (53.1)	0.101
Antral follicle count in left ovary	5.45 ± 3.17	5.08 ± 2.91	0.113
Antral follicle count in right ovary	5.59 ± 3.05	6.09 ± 2.73	0.022
Anti-Müllerian hormone (AMH)	2.25 ± 1.99	2.53 ± 2.14	0.076
Baseline Sex Hormone			
Follicle-Stimulating Hormone (FSH)	9.26 ± 5.85	10.02 ± 8.48	0.177
Luteinizing Hormone (LH)	4.42 ± 5.06	4.95 ± 7.99	0.313
Estradiol (pg/mL)	63.11 ± 63.32	62.69 ± 46.50	0.922
Total Testosterone (ng/mL)	0.45 ± 0.32	0.46 ± 0.48	0.815

**Table 2 jcm-11-06395-t002:** Outcomes of controlled ovarian hyperstimulation.

Parameters	FET (n = 345)	Fresh ET (n = 375)	*p*-Value
Total amount of gonadotropins (IU)	2452.70 ± 1054.21	2479.18 ± 1022.14	0.734
Days of ovarian stimulation	10.43 ± 3.59	10.59 ± 3.33	0.546
Estradiol level on day of HCG (pg/mL)	3088.90 ± 2625.68	2812.09 ± 1940.05	0.108
Progesterone level on day of HCG (ng/mL)	1.39 ± 1.44	1.26 ± 0.80	0.152
Endometrial thickness before transfer (mm)	9.67 ± 1.81	9.85 ± 2.12	0.226
Treatment Protocols			
Depot GnRHa protocol	199 (57.7)	197 (52.5)	0.165
Flexible short protocol	146 (42.3)	178 (47.5)	0.165
Number of oocytes retrieved	8.12 ± 6.78	8.16 ± 4.79	0.919
Insemination Modes			
IVF, n (%)	285 (82.6)	304 (81.1)	0.592
ICSI, n (%)	60 (17.4)	71 (18.9)	0.592
Number of good quality embryos	1.9	1.92	0.873
Number of embryos transferred	1.83 ± 0.37	1.80 ± 0.39	0.321

Abbreviations: HCG—human chorionic gonadotrophin, IVF—in vitro fertilization, ICSI—intracytoplasmic sperm injection.

**Table 3 jcm-11-06395-t003:** Fertility outcomes and neonatal outcomes after FET cycles and fresh ET cycles.

Parameters	FET (n = 345)	Fresh ET (n = 375)	*p*-Value
Fertility Outcome			
Biochemical pregnancy, n (%)	117 (33.9)	143 (38.1)	0.239
Clinical pregnancy, n (%)	91 (26.4)	126 (33.6)	0.035
Implantation rate (%)	15.2 (96/631)	19.7 (133/675)	0.265
Ectopic pregnancy, n (%)	3 (0.9)	2 (0.5)	0.587
Miscarriage, n (%)	20 (5.8)	32 (8.5)	0.156
Live birth, n (%)	68 (19.7)	91 (24.3)	0.141
Neonatal Outcomes			
Birth weight (g)	3217.16 ± 734.44	3003.37 ± 635.00	0.037
Low birth weight among live newborns, n (%)	10 (12.3)	17 (16.8)	0.397
Macrosomia, n (%)	5 (6.2)	5 (5.0)	0.753
Congenital anomalies among live newborns	1 (1.2)	0	0.445
Perinatal mortality	0	0	

Low birth weight was defined by birth weight less than 2500 g. Macrosomia was defined by birth weight more than 4000 g.

**Table 4 jcm-11-06395-t004:** Maternal outcomes of controlled ovarian hyperstimulation.

Maternal Outcome	FET (n = 345)	Fresh ET (n = 375)	*p*-Value
Moderate or severe ovarian hyperstimulation syndrome	0	1 (0.3)	1.000
Gestational hypertension	1 (1.1)	2 (1.6)	1.000
preeclampsia	3 (3.3)	3 (2.4)	0.697
HELLP syndrome	0	0	
Gestational diabetes	3 (3.3)	5 (4.0)	1.000
PPROM	4 (4.4)	7 (5.6)	0.765
preterm birth, n (%)	9 (2.6)	21 (5.6)	0.046
Cesarean section, n (%)	45 (13.0)	62 (16.5)	0.188

Abbreviations: HELLP syndrome—hemolysis, elevated liver enzymes, and low platelet count syndrome, PPROM—preterm premature rupture of the membranes.

## Data Availability

Data are contained within the article.

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
