# Peer review of "Pregnancy Outcomes after Frozen Embryo Transfer and Fresh Embryo Transfer in Women of Advanced Maternal Age: Single-Center Experience"

_jcm, 2022, doi:10.3390/jcm11216395_

Round 1

Reviewer 1 Report

This is an interesting paper,  however few information need to be added to give a better understanding for the reader.

1. Medical indication for freeze all patients

2. Freeze all embryo are vitrified at d3 post insemination.  In what stage the embryos were transfered on frozen ET and fresh ET. Could you please provide these information in teh manuscript.

3. If possible vitrification and warming protocols

Reviewer 2 Report

The article talks about the effect of FER in AMA. tries to highlight if for AMA patients they give more pregnancies in fresh or thaw cycles. statistics and data are well reported. certainly the topic is very much felt in these times. It should be noted that in the AMA the fresh transfer gives greater results and fewer problems than the defrosted transfer. However, the difference between the IVF Vs ICSI insemination techniques is not reported and in the literature the debate also focuses on the technique used. In fact, it is not clear if the insemination method is a cofactor of the related problems or if the problems on the weight of the births, on the days of gestation is due to the fertilization technique, to the in vitro development or to the biology of the patients underlying ART. still the "long" protocols in AMA patients is not the most suitable, to date the literature especially in AMA patients use protocols with Antagonist and triggering with the Agonist. In fact, the average number of oocytes in the two groups is always 6. most likely having few follicles and aspirating them all minutely, the fresh transfer is the best strategy avoiding the stress of cryopreservation to the same embryos and negative hormonal feedback due to a high number of follicles that luteinize produce progesterone. However, the work well describes the state of the art in AMA patients, highlighting that pregnancy rates in fresh cycles are better in AMA patients in the first IVF cycle.

Reviewer 3 Report

Current article. Important questions are identified regarding the development of individual and  most appropriate  protocols depending on the age and ovarian response in women with AMA. The authors obtained very important results that require confirmation in the course of multicenter studies.

I have no fundamental remarks

Reviewer 4 Report

General comments

This retrospective study used enough samples to gain insight in the result of ET and FET. In terms of the protocol in embryo production, retrieval and patient preparation are the routine and standard of an infertility clinic. However, in my opinion, additional information needs to be added in the following part.

1.      For the embryo vitrification protocol, please give a brief explanation in this part. Add after line 103

2.      The quality of the embryo after thawing needs to be explained. Which quality that eligible to be transfer? The criteria should be well defined. Add somewhere after lines 110-111.

3.      In the discussion part. Since the AMA factor (age of FE vs. FET is the same), the possible factor influenced is the difference between FE vs. FET embryo. Regarding that issue, I didn't see any discussion about the comparison between FE vs. FET embryos in terms of embryo quality, and its viability for embryo transfer. Then there is also no explanation about the effect of frozen protocol on embryos. As a suggestion, please add those issues and the related ones. The explanation can be retrieved from an earlier study on the different embryos of mammals.

Reviewer 5 Report

"Pregnancy outcomes after frozen embryo transfer and fresh embryo transfer in women of advanced maternal age: Single-center experience” by Yao Chen, Yandong Chen, Jihong Yang, Yingying Hao, Ting Fei, Ruizhi Feng and Yun Qian.

Let me start by declaring no potential or perceived competing interests that may influence my review.

In comparison to patients receiving their first fresh embryo transfer, the authors sought to compare the maternal and newborn outcomes of patients receiving their first frozen embryo transfer following a freeze-all cycle. The success rates of frozen and fresh embryo transfers as well as their advantages have been hotly contested topics. Many fertility specialists claim that using frozen embryo transfers via assisted reproductive technology (ART) results in a higher pregnancy success rate than using fresh embryos. Choosing a fresh embryo transfer versus a frozen one depends on many circumstances. Success rates, nevertheless, are not the only deciding element, but are an important decision element throughout the ART process. Furthermore, it's important to distinguish between opposing findings when it comes to older women.

 The authors should address the following concerns:

 A careful reading of the manuscript must be carried out in order to avoid missing spaces between words and to correct minor syntax errors.

 Authors’ affiliations:

Please clarify the distinction between affiliations 2 and 3.

 Abstract section:

The authors state that FET resulted in higher birthweights and was associated with a lower incidence of preterm births. Please provide the incidence of preterm births.

Explain the conclusion about the freeze-all strategy benefiting high responders but not AMA women when the manuscript is about AMA

 Introduction section:

Lines 42-43: Please provide a reference for this statement.

 Lines 49-50: “However, FET had a significantly higher rate of large for gestational age, macrosomia, and several maternal complications [10-12].” Please rephrase this sentence.

 Material and Methods section:

Please provide the number of trial approval by the ethics committee.

 Define your interval/range of AMA for both groups.  

 What were the reasons for freezing all the embryos?

 AMA was at the time of transfer, at the time of IVF plus freezing, or both? Please clarify.

 In the legend of Figure 1 please provide the abbreviations.

 In figure 1, by “transfer of cleavage stage embryos were included” what do authors mean? Please clarify.

 Do not use didn’t instead of did not.

 How many women enrolled in fresh and frozen embryo transfer were stimulated by protocol 1 or 2? Please provide these numbers.

 In each group of women, how many were enrolled in IVF or ICSI cycles? This must be clarified and the statistics take this variant into account.

 Please provide the embryo grade system used.

 Results section:

 Line 145: Please define the abbreviation AFC.

 Table 1: Please provide the units for duration of infertility.

 According to table 1, the women enrolled were already mother. Were they all? How many conceived previously naturally and how many resorted to ART?

 Discussion section:

Line 227: What do authors mean by good outcomes? Please clarify.

 Lines 233-237: It is significant these some patients? Why had they poorest prognosis? How low were their oocyte yield or premature luteinisation? And how was these compared to women with fresh oocyte transfer?

 Lines 257-259: Please provide a reference for your statement.

 Lines 262-263: How old was the only patient diagnosed with moderate OHSS in the fresh ET group?

 Line 283: Please provide a reference for this sentence.

 Conclusions section:

Define high, intermediate and low responders in material and methods

 References section

Please follow the guidelines of the journal as the example:

Capitalize and/or abbreviate the name of the journal in references 4, 10, 12, 14, 19, 20, 27, 28 and 29.

Round 2

Reviewer 4 Report

the paper has been revised as suggested in the previous comments

Author Response

Point1: the paper has been revised as suggested in the previous comments.

Response1: Thank you for your comments.